# The Clinical Application of Immunohistochemical Expression of Notch4 Protein in Patients with Colon Adenocarcinoma

**DOI:** 10.3390/ijms24087502

**Published:** 2023-04-19

**Authors:** Marlena Brzozowa-Zasada, Adam Piecuch, Marek Michalski, Natalia Matysiak, Marek Kucharzewski, Marek J. Łos

**Affiliations:** 1Department of Histology and Cell Pathology in Zabrze, Faculty of Medical Sciences in Zabrze, Medical University of Silesia in Katowice, 40-055 Katowice, Poland; 2Silesian Nanomicroscopy Centre in Zabrze, Silesia LabMed- Research and Implementation Centre, Medical University of Silesia, 40-055 Katowice, Poland; 3Faculty of Health Sciences, Jan Dlugosz University of Czestochowa, 42-200 Czestochowa, Poland; 4Department of Pathology, Pomeranian Medical University, 71-344 Szczecin, Poland; 5Autophagy Research Center, Shiraz University of Medical Sciences, Shiraz 7134845794, Iran

**Keywords:** Notch4, colon adenocarcinoma, 5-year survival rate, immunohistochemistry (IHC), immunogold labelling

## Abstract

The Notch signalling pathway is one of the most conserved and well-characterised pathways involved in cell fate decisions and the development of many diseases, including cancer. Among them, it is worth noting the Notch4 receptor and its clinical application, which may have prognostic value in patients with colon adenocarcinoma. The study was performed on 129 colon adenocarcinomas. Immunohistochemical and fluorescence expression of Notch4 was performed using the Notch4 antibody. The associations between the IHC expression of Notch4 and clinical parameters were analysed using the Chi^2^ test or Chi^2^_Yatesa_ test. The Kaplan–Meier analysis and the log-rank test were used to verify the relationship between the intensity of Notch4 expression and the 5-year survival rate of patients. Intracellular localisation of Notch4 was detected by the use of the immunogold labelling method and TEM. 101 (78.29%) samples had strong Notch4 protein expression, and 28 (21.71%) samples were characterised by low expression. The high expression of Notch4 was clearly correlated with the histological grade of the tumour (*p* < 0.001), PCNA immunohistochemical expression (*p* < 0.001), depth of invasion (*p* < 0.001) and angioinvasion (*p* < 0.001). We can conclude that high expression of Notch4 is correlated with poor prognosis of colon adenocarcinoma patients (log-rank, *p* < 0.001).

## 1. Introduction

Colorectal cancer (CRC) is the third most common cancer worldwide [1]. Important factors associated with the development of this type of cancer include drinking alcohol, smoking, unhealthy dietary patterns and obesity [2]. Among factors which are clearly correlated with disease development, it should also be mentioned about ageing, genetic mutations and hereditary factors [3]. The development of colorectal cancer occurs gradually as a sequence of specific morphological and genetic changes. One of the most common types of colorectal cancer is adenocarcinoma (COAD). This type of cancer develops from colorectal glandular epithelial cells. Under pathological conditions, the shape of epithelial cells changes and grows out of control, leading to the development of adenoma and adenocarcinoma [4].

With the increase in colorectal cancer screening, a significant reduction in the incidence of COAD has been observed; however, the mortality is expected to reach 60% by 2035 [5]. It should be noted that despite advances in currently available standard treatments, e.g., surgery, chemotherapy, radiotherapy and immunotherapy, the 5-year overall survival (OS) of patients with COAD remains poor [5,6]. Therefore, there is an urgent need for the development of novel biomarkers to improve the outcome of patients, allowing earlier therapeutic intervention and reducing the increasing burden of COAD [7,8].

Notch signalling is one of the conserved and well-characterised signalling systems involved in tissue homeostasis and the development of many diseases, including cancer [9]. There are four Notch receptors (Notch1-4) and five Notch ligands (Delta-like (DLL) 1, 3, 4 and Jagged (JAG) 1–2) in mammals. The Notch full-length receptor is initially cleaved (S1 cleavage), and after this, it undergoes post-translational modification (glycosylation) in the Golgi apparatus. From there, it is transported into the plasma membrane as a transmembrane heterodimeric protein [10,11]. In the canonical Notch signalling pathway, the Notch receptors are proteolytically cleaved by the ADAM10 disintegrin and metalloprotease (ADAM10) domain (S2 cleavage) as well as the γ-secretase complex (S3 cleavage), leading to the release of the Notch intracellular domain (NICD). This domain enters the nucleus and binds to the DNA-binding protein CSL (CBF-1 (RBPJ)/suppressor of hairless/Lag1), which recruits mastermind-like protein (MAML) to activate transcription of Notch target genes such as the *Hes* and *Hey* families [12].

Some studies revealed that cancer stem cell self-renewal, epithelial to mesenchymal transition (EMT), and radio- or chemotherapy resistance might be related to mutations and amplifications of the Notch ligand or/and Notch receptors [9,10,12]. Among them, it is worth noting the Notch4 receptor. The high expression of this protein was found in such cancers as hepatocellular carcinoma (HCC) [13,14,15], intrahepatic cholangiocarcinoma [16], melanoma [17,18,19], oral squamous cell carcinoma (OSCC) [20,21], breast carcinoma (BC) [22,23,24,25], gastric carcinoma [26], non-small cell lung carcinoma (NSCLC) [27,28,29] and acute myeloid leukaemia (AML) [30,31]. In patients with colorectal cancer, Shaik et al. demonstrated that expression of Notch4 was significantly higher than in healthy individuals or patients with adenoma [32]. Similar results have been obtained in the Chinese population [33,34]. Nevertheless, we have little data on the clinical application of Notch4 protein expression in European patients with colorectal cancer, especially with colon adenocarcinoma. With this in mind, we decided to investigate the expression of Notch4 protein in European (Polish) patients with colon adenocarcinoma without any therapy prior to radical surgery. Moreover, we also investigated the association between Notch4 protein expression and the clinicopathological factors. The prognostic activity of Notch4 protein was analysed in relation to 5-the year survival of patients, which is a very significant factor from the clinical oncology point of view. In view of the Notch signalling pathway and the stages of formation of both precursors and active Notch receptors, it can be hypothesised that Notch4 expression is associated with the cell membrane, the nuclear membrane and matrix, and the Golgi apparatus. Unfortunately, studies on the localisation of the Notch4 receptor within the cells, especially in cancerous cells, are rare. One of these studies assesses the intranuclear and intranucleolar localisation of Notch4 in breast cancer cells [35]. Therefore, the aim of our research was also to determine Notch4 intracellular localisation within the cells of colon adenocarcinoma tissue by the use of the immunogold labelling method and TEM.

## 2. Results

### 2.1. Patients’ Characteristics

The study included 129 colon adenocarcinomas. Among patients, there were 67 men and 62 women (mean age: 65 years; range: 56–77 years). In 66 (51.16%) cases, the tumours were situated in the right colon and the left colon in 63 (48.84%). The three histological differentiation levels were used for classification, which were as follows: G1 (well-differentiated cancers where cells are similar to healthy cells)—25 cases (19.38%), G2 (moderately differentiated where cells are somewhat like healthy cells)—66 cases (51.16%) and G3 (poorly differentiated where the cells are not similar to healthy colonocytes)—38 (29.46%). (Table 1). Among the adenocarcinomas, 15 (11.63%) were at T1 depth of invasion (the tumour has grown into the submucosa), 18 (13.95%) were at T2 (The tumour has grown into the muscularis propria), 73 (56.59%) were at T3 (the tumour has grown through the muscularis propria and into the subserosa), and 23 (17.83%) were at T4 (the tumour has grown into the surface of the visceral peritoneum).

In samples of colon adenocarcinoma, the positive immunohistochemical reaction indicating the presence of Notch4 protein was observed in the cytoplasm and nucleus of cancer cells. The positive reaction was also detected within the cells of healthy colon tissue (Figure 1). It is important to mention that expression was described as strong in the vast majority of colorectal adenocarcinoma tissues, whereas expression in cells of healthy surgical margins was determined to be low.

### 2.2. Correlations between Notch4 Immunohistochemical Expression and Clinicopathological Parameters

Among the study cohort, 101 (78.29%) colon adenocarcinoma samples showed a high level of immunohistochemical expression of Notch4 protein, whereas only 28 (21.71%) demonstrated a low level of immunoreactivity. The immunohistochemical status of Notch4 was correlated with the clinicopathological features of patients and the 5-year survival rate. The level of Notch4 expression was found to be significantly related to the histological grade of the tumour (*p* < 0.001, Chi^2^ test). Notch4 protein expression was found to be high in 6 (24.00%), 58 (87.88%), and 37 (97.37%) of G1, G2, and G3 tumours, respectively. In contrast, the low level of immunohistochemical expression of notch4 protein was found in 19 (76.00%), 8 (12.12%), and 1 (2.63%) of G1, G2, and G3 tumours, respectively. Furthermore, Notch4 expression was associated with the expression of PCNA antigen (*p* < 0.001, Chi^2^_Yatesa_ test). Notch4 protein was found to be highly expressed in 5 (23.81%) and 96 (88.89%) samples with low and high levels of PCNA immunoreactivity, respectively (Table 2; Figure 2).

It is worth noting that Notch4 expression was also related to angioinvasion (*p* < 0.001, Chi^2^_Yatesa_ test). High Notch4 immunohistochemical expression was found in 89 (89.90%) of the patients with no angioinvasion, while low immunoreactivity was found in 10 (10.10%) patients. In contrast, 12 (40.00%) patients with angioinvasion had high Notch4 expression, whereas 18 (60.00%) patients had low Notch4 immunoreactivity. Notch4 immunohistochemical expression was also related to the depth of invasion (*p* < 0.001, Chi^2^_Yatesa_ test). For those patients who were characterised as T1/T2, a high level of immunohistochemical reaction was noted in 18 (54.55%) and a low level of expression was detected in 15 (45.45%). For T3/T4 patients, a strong Notch4 immunohistochemical reaction was reported in 83 (86.46%) patients, while low expression was detected in 13 (13.54%) (Table 3).

### 2.3. Prognostic Role of Notch4 Expression in Colon Adenocarcinoma

The prognostic significance of Notch4 expression in colon adenocarcinoma patients was analysed in relation to a 5-year survival rate. All samples were assessed by Kaplan–Meier survival curves. The 5-year survival rate was significantly higher in the group of patients where a low level of Notch4 expression was found (log-rank, *p* < 0.001) (Figure 3).

Additionally, the value of Notch4 expression in the context of the 5-year survival rate was evaluated in patients’ subgroups stratified by grade of histological differentiation, depth of invasion, staging and PCNA expression (Figure 4). Interestingly, the expression of Notch4 was not related to the 5-year survival rate in patients stratified according to G1 (log-rank test, *p* = 0.412, G2 (log-rank test, *p* = 0.181) and G3 (log-rank test, *p* = 0.007). In contrast, in the group of patients with T1/T2 depth of invasion, patients with a low level of Notch4 immunohistochemical reaction showed significantly higher 5-year survival in comparison to patients with high expression of this protein (log-rank test, *p* < 0.011). Similar results have been obtained in patients with T3/T4 depth of invasion (log-rank test, *p* = 0.001). Moreover, in patients with stage I of the disease, the low expression of Notch4 was also associated with the 5-year survival rate (log-rank test, *p* < 0.001). In patients with stage III, low expression of Notch4 was also associated with a 5-year survival rate, although these results were not statistically significant (log-rank test, *p* = 0.052). Interestingly, the expression of Notch4 was associated with the 5-year survival rate of patients with a high level of PCNA expression. The patients with a high level of this antigen and a low level of Notch4 expression has significantly higher 5-year survival rate (log-rank test, *p* = 0.006).

Univariate Cox regression analyses revealed that Notch4 immunohistochemical expression, histological grade, depth of invasion, angioinvasion and expression of PCNA were significant prognostic factors. Multivariate analysis found that, in our cohort of patients, the degree of histological differentiation and Notch4 expression is regarded as independent predictors related to 5-year survival in patients with colon adenocarcinoma (Table 4).

### 2.4. Immunofluorescence Staining

Based on the study by Frithiof et al. [36], we wanted to check the expression of Notch4 in colon adenocarcinomas using immunofluorescence. Therefore, we randomly selected 50 slides with tissue sections treated with anti-Notch4 antibody and Dako Liquid Permanent Red (10 controls, 25 described previously as low expression by the use of IHC and 25 described previously as high expression by the use of IHC). However, it is important to point out that we used this technique as a supplementary one. Nevertheless, the results obtained are very promising and suggest that tissue sections stained with anti-Notch4 antibody and treated with LPR chromogen can be used for immunofluorescence analysis. The intensity of Notch4 expression in both non-neoplastic tissue and tumour tissue was determined using software Zen 2 (blue edition). A fluorescent signal (red colour signal) of varying intensity was found in cells of non-neoplastic mucosa and cancer cells. In some cancer cells, the expression and fluorescence signal was found in the cytoplasm of the apical parts of the cells, while in others, intense fluorescence was found throughout all cytoplasm of the cells or in the cell nuclei (Figure 5).

### 2.5. Intracellular Localisation of Notch4 by the Method of Immunogold Labelling with the Use of Transmission Electron Microscopy (TEM)

The immunogold labelling method was used to reveal the localisation of Notch4 protein at the cellular level within colorectal adenocarcinoma tissues and in non-neoplastic cells from surgical margins. In non-neoplastic cells, electron-dense granules were detected in close proximity to the cellular membrane and in the apical part of the cells. In cancer cells, black granules were found within the cisterns of the rough endoplasmic reticulum and in mitochondria. In some cancer cells, granules indicating the presence of Notch4 were visible within the nuclei. (Figure 6). In the fibroblasts of non-pathological colon tissue, Notch 4 expression was found in the cell membrane, the nuclear membrane and the endoplasmic reticulum. Images showing the immunocytochemical localisation of Notch4 in fibroblasts are shown in the Appendix A.

## 3. Discussion

The Notch signalling pathway was first characterised to play an oncogenic role in T-cell Acute lymphoblastic leukaemia (T-ALL) [37]; however, the similar role of the Notch receptors and Notch ligands have been found in breast cancer [21,22,23], lung adenocarcinoma [26,27,28], hepatocellular carcinoma [12,13,14] and ovarian cancer [38,39]. The potential mechanisms of carcinogenesis associated with the oncogenic activity of Notch involve such biological events as the control of the phenotype of cancer-initiating cells, upregulation of tumour-associated signalling factors such as P53, facilitation of tumour angiogenesis and invasion, and cell cycle regulation [8,9,10]. It should also be noted that Notch may function as a tumour suppressor in other cancers, like squamous cell carcinoma (SCC) and neuroendocrine tumours [40]. The anti-tumour activity is related to the regulation of the malignant transcription factors, downstream suppressor gene activation and suppression of the cell cycle [8,9,10].

A great number of studies have addressed the role of Notch4 in cancer, particularly the molecular mechanisms associated with it. The majority of studies have suggested that Notch4 expression is upregulated during the development of cancer. Moreover, this receptor is also known to be involved in the regulation of stem cell-like self-renewal, epithelial-mesenchymal transition (EMT), radio/chemoresistance and angiogenesis. Interestingly, the expression level of Notch4 is different in different types of tumours [11]. Results of our study demonstrated that expression of Notch4 in colon adenocarcinoma tissue was clearly upregulated in comparison to that observed in the healthy tissue of the surgical margin. The results that we obtained using immunohistochemistry and immunofluorescence techniques revealed that Notch4 expression is associated with the cytoplasm and cell nucleus. Furthermore, by the use of the immunogold labelling method, we have confirmed the Notch4 presence in the cytoplasm and nucleus of tumour cells. Notch4’s nuclear and nucleolar localisation has been found by Saini et al. in breast cancer cells. Probably, it can stabilise the DNA repair machinery, thus allowing the recovery of cells under genotoxic stress damage [35]. In colon adenocarcinoma cells, the black granules indicating the presence of Notch4 antigen were localised in the cytoplasm. They were detected mostly in the vicinity of membranous organelles, including the endoplasmic reticulum and mitochondria. Tumour cells are marked by the fact that different signalling pathways are altered in relation to healthy cells. Moreover, these cells reside in a tumour-specific microenvironment where a network of interactions exists. Perhaps as a result of a disrupted protein transport system, a feature that is quite common in cancer cells, this protein may have been displaced into the mitochondria, as can be seen in the images showing its immunocytochemical localisation within the cancer cells [9,10,11]. It is also possible that the Notch signalling pathway itself has been disrupted, and these pathway proteins have been incorrectly directed to other organelles, including the endoplasmic reticulum or mitochondria [9].

It should be mentioned that in our cohort of patients, approximately 78% of colon adenocarcinoma specimens demonstrated high Notch4 protein expression, while low levels of immunoreactivity were found only in 22% of cases. High Notch4 expression was markedly correlated with the histological grade of the tumour (*p* < 0.001, Chi^2^ test), depth of invasion (*p* < 0.001, Chi^2^_Yatesa_ test), angioinvasion ((*p* < 0.001, Chi2Yatesa test) and PCNA immunohistochemical expression (*p* < 0.001, Chi^2^_Yatesa_ test). Interestingly, the strong expression of Notch4 protein was noted in 24% of G1 tumours, 88% of G2 tumours and 97% of G3 tumours. These results may indicate that Notch4 plays an important role in colon adenocarcinoma progression and may be an identification biomarker for patients with a more aggressive form of this malignancy. In this context, it is worth noting that Notch4 expression has also been associated with PCNA immunohistochemical expression (*p* < 0.001, Chi^2^_Yatesa_ test). The high level of Notch4 reactivity was revealed in 24% and 89% of samples with low and high PCNA expression, respectively. PCNA, a non-histone nuclear protein, has a molecular mass of 36 kDa and is a specific marker of cell division. Its action is associated with DNA polymerase, synthesised shortly before the S-phase of the cell cycle. However, this protein is also connected with the machinery associated with the DNA repair mechanism [41]. Nevertheless, this protein could be as important as ki67 in terms of prognosis. During the planning of our research, we decided also to examine the value of Notch4 expression in terms of 5-year survival in a cohort of patients who were stratified according to low and high levels of expression of PCNA. In patients with high levels of PCNA expression, there was a statistically significant difference in estimated survival time between those with high and low levels of Notch4 expression (log-rank, *p* = 0.006). For example, patients with low expression of PCNA had a median survival rate of 45 months, whereas the patients with high expression had a median survival rate of 24 months. Evaluation of PCNA and Notch4 expression may therefore have a significant clinical relevance by indicating patients who may have a significantly worse prognosis. Similarly, patients in the T1/T2 group showed a significant statistical difference in survival time. Patients characterised by a high level of Notch4 expression had a significantly lower survival time than the group with a low level of Notch4 expression (log-rank, *p* < 0.001). Similar results were obtained in the group of T3/T4 (log-rank, *p* = 0.010).

In the context of our study, interesting results have been obtained by Ahn et al., who demonstrated that in patients with hepatocellular carcinoma, the high expression of Notch4 is correlated with low Edmondson grade, low AJCC T-stage, lack of microvascular invasion, absence of intrahepatic metastases and low serum AFP levels [12]. In contrast, in patients with intrahepatic cholangiocarcinoma, the high Notch4 expression correlates with high serum CA125 levels [15]. In patients with oral squamous cell carcinoma, the high expression of Notch4 is correlated with poor differentiation, advanced clinical stage, periosteal invasion and lymph node metastasis [18,19]. Qian et al. demonstrated that the activation of Notch4 was related to the induction of gastric cancer growth in vitro and in vivo, while Notch4 inhibition using Notch4 siRNA had opposite effects [42]. In patients with Non-Small Cell Lung Cancer (NSCLC), Notch4 expression was positively associated with tumour size, lymph node metastasis (LNM), distal metastasis (DM), and depth of invasion (T). Patients with a high level of this protein had significantly lower OS than patients with a low level of Notch4 expression [26,27,28]. Probably the poor clinical outcome of cancer patients with high expression of Notch4 is associated with its role in the mechanism of EMT, which is a very significant molecular event leading to cancer metastasis. Zhang et al. revealed that activation of Notch4 signalling, which is dependent on the activation of NF-kB, promotes the growth, metastasis, and EMT of tumour cells in prostate cancer [43]. In melanoma and head and neck squamous cell carcinoma, Notch4 signalling induces EMT by stimulating the expression of EMT markers such as Vimentin and Twist1 and downregulating the expression of E-cadherin [17,44]. In this place, it is worth noting the role of Notch4 in melanoma. In this cancer, upregulation of Notch4 expression promotes metastasis through the regulation of Twist1 expression, which indicates a poor prognosis. Nevertheless, others have reported that high Notch4 expression enhances the expression of E-cadherin and attenuates melanoma malignant behaviour. Importantly, Notch4 may induce suppression of Snail2 and Twist1 through downstream *Hey1* and *Hey2* targets and is non-canonically mediated in WM9 and WM164 melanoma cell lines [18]. Probably, the poor clinical prognosis in high Notch4 patients might be related to vascular mimicry (VM), which is a tumour microcirculation system imitating the layout of the embryonic vascular network to provide oxygen and nutrients to tumour cells and, importantly, is epithelium independent [45]. The expression of Notch4 and VM has shown a positive connection in the case of NSCLC and HCC patients [27]. Bao et al. revealed that the oncogenic circular RNA 7 stimulated Notch4 expression in HCC, enhancing VM development and inhibiting miR-7-5p expression in HCC [14].

It may be beneficial to suppress Notch4 signalling in cancer because Notch4 is frequently recognised as a crucial participant in oncogenesis. The varied functions of Notch4 signalling in cancer, as well as the possible outcomes and clinical utility of applying multiple Notch4-targeting treatment techniques, are likely to be further clarified in studies.

## 4. Materials and Methods

### 4.1. Patients and Tumour Samples

Tissue colon material collected from the patients undergoing colon resection at the Municipal Hospital in Jaworzno between January 2014 and December 2015 with histopathologically confirmed colon adenocarcinoma was used for the study. Patients who received preoperative radiotherapy or chemotherapy, patients with serious complications or distant metastasis, patients undergoing resection from tumour recurrence, patients with adenocarcinoma in the setting of inflammatory bowel disease and patients with histopathologically confirmed subtype other than adenocarcinoma were excluded from the study. Based on an established protocol, histopathological sections containing tumour fragments and adjacent tissue sections without tumour lesions were taken from each surgical specimen. The collected samples were fixed in formalin and embedded in paraffin blocks. In the next step, the paraffin blocks were cut, and sections were routinely stained with H&E to confirm the histopathological diagnosis. Sections containing tissue margins were also assessed. If tumour cells were found, the material was excluded from the study. To determine whether Notch4 protein had prognostic significance, patients were followed up for 5 years to estimate the 5-year survival rate.

### 4.2. Immunohistochemical and Immunofluorescence Staining

Paraffin-embedded tissue blocks with formalin-fixed colon adenocarcinoma specimens and resected margins were cut into 4-µm-thick sections, fixed on Polysine slides and deparaffinised in xylene and rehydrated through a graded series of alcohol. To retrieve the antigenicity, the tissue sections were treated with microwaves in a 10 mM citrate buffer (pH 6.0) for 8 min each. Subsequently, sections were incubated with antibodies to Notch4 (GeneTex. polyclonal antibody. Cat. No. GTX03453, final dilution 1:600, Irvine, CA, USA), which targeted cleaved N-terminous epitope, and PCNA (GeneTex. polyclonal antibody. Cat. No. GTX100539, final dilution 1:600, Irvine, CA, USA). For visualisation of protein expression, the sections were treated with BrightVision (Cat. No. DPVB55HRP WellMed BV, ’t Holland 31, 6921 GX Duiven, The Netherlands) detected system and Permanent AP Red Chromogen (Dako LPR from Agilent Technologies Code K0640). Mayer’s haematoxylin was used to counterstain the nuclei. In addition, the expression of Notch4 and PCNA was studied in sections of healthy mucosa from patients undergoing screening colonoscopy with no inflammatory or cancerous lesions. For the analysis of the results of the immunohistochemical staining, we have adapted the immunoreactive score on the basis of previous publications [46,47]. The scoring of Notch4 expression and PCNA expression was based on both the intensity and extension of immunohistochemical reaction determining the presence of Notch4. The intensity was graded as follows: 0, no signals; 1, weak; 2, moderate; and 3, strong staining. The frequency of positive cells was determined semiquantitatively by assessing the whole section, and each sample was scored on a scale of 0 to 4: 0, negative; 1, positive staining in 10–25% cells, 2, 26–50% cells; 3, 51–75% cells; and 4, 76–100% cells. A total score of 0–12 was finally calculated and graded as; I, score 0–1; II, 2–4; III, 5–8; IV, 9–12. Grade I was considered negative, and grades II, III and IV were positive. Grades I and II represented no or weak staining (low expression), and grades III and IV represented strong staining (high expression).

The evaluation was carried out by two independent pathologists. Differences were again assessed until consensus was obtained.

Additionally, tissue sections treated with anti-Notch4 antibody and Dako Liquid Permanent Red (LPR) were visualised with a confocal fluorescent microscope (Zeiss LSM 980 with Airscan 2; Zeiss; Germany). LPR fluorescence representing Notch4 protein was visualised with 592 nm excitation and 574–735 nm emission using TexRed filters sets. The intensity of Notch4 expression in both non-neoplastic tissue and tumour tissue was determined using software Zen 2 (blue edition) (Zeiss; Germany).

### 4.3. Immunogold Electron Microscopy

For the study with the use of immunogold labelling methods, the selected areas of non-neoplastic colon tissue from surgical margins and samples of colon adenocarcinomas (10 patients) were fixed in 4% paraformaldehyde in 0.1 M phosphate-buffered saline (PBS) for 2 h at room temperature and then washed several times in PBS. After washing, the specimens were dehydrated in a graded ethanol series and infiltrated in a 2:1 (*v*:*v*) ethanol/LR White mixture and 1:2 (*v*:*v*) for 30 min each on ice. Afterwards, the samples were infiltrated in pure LR White Acrylic resins (Sigma Aldrich Cat. No. L9774). Ultra-thin sections (70 nm) were cut with a RMC Boeckeler Power Tomo PC ultramicrotome with a diamond knife (45°; Diatom AG, Biel, Switzerland). Ultrasections were mounted on 200-mesh nickel grids coated with Formvar and immunolabelled. Sections on the grids were preincubated first for 30 min by floating on drops of 50 mM NH4 Cl in PBS and subsequently blocked for 30 min on drops of 1% BSA in PBS. The grids were then incubated overnight (16–18 h) at 4 °C with primary anti-Notch4 antibody (GeneTex. polyclonal antibody. Cat. No. GTX03453) diluted 1:20 in BSA. The bound antibodies were localised by incubating the sections for 1 h on Immunogold-conjugated goat anti-mouse IgG 15 nm (BBInternational BBI Solutions, Sittingbourne, UK) diluted 1:100. Lastly, the grids were washed on PBS drops (five changes, 5 min each) and water (three changes, 3 min each) before staining with 0.5% aqueous uranyl acetate. In controls, the primary antibody was not used. The grids were then air-dried and analysed in a TECNAI 12 G2 Spirit Bio Twin FEI Company transmission electron microscope at 120 kV. Images were captured using a Morada CCD camera (Gatan RIO 9, Pleasanton. CA, USA).

### 4.4. Statistical Analysis

The associations between the IHC expression of Notch4 and clinical parameters were analysed statistically with Statistica 9.1 (Software, StatSoft, Cracow, Poland). All the quantitative variables were described as medians and ranges. The Chi^2^ test and Chi^2^_Yatesa_ test were used to compare the analysed groups. The Yates correction was applied to 2 × 2 tables when at least one of the boxes had an expected count of less than 10.

Kaplan–Meier analysis and the log-rank test were used to verify the relationship between the intensity of Notch4 expression and 5-the year survival rate of patients. The results were considered statistically significant when *p* < 0.05. The correlation of signal intensity indicating the presence of Notch4 protein between the different groups, i.e., non-pathological tissue and in tissues previously identified by IHC as low expression and high expression, was determined by an ANOVA test.

## 5. Conclusions

Based on the results obtained in the Cox regression model, Notch4 has been identified as a protein connected with the reduced 5-year survival of colon adenocarcinoma patients. The multivariate analysis revealed that the grade of histological differentiation and immunohistochemical expression of Notch4 in colon adenocarcinoma tissue could be considered independent prognostic factors.

In this place, it should be pointed out that our study is the first which demonstrate immunohistochemical expression of Notch4 in colon adenocarcinoma tissues in patients from European populations. Furthermore, it also reveals the prognostic value of Notch4 expression in patients stratified along certain criteria that are relevant from a clinical oncology point. In this case, PCNA expression level, depth of invasion (T-value) and angioinvasion were taken into account. Our work is also the first to show the localisation of Notch4 in tumour tissue at the electron microscopic level by using the immunogold labelling method and confocal fluorescence microscope.

Nevertheless, our study has some limitations that need to be mentioned. The size of the studied cohort was limited, and the patients came from a single hospital, which may introduce a selection bias into the study. Future studies should be conducted to increase the sample size and in vitro experiments to understand the mechanism of Notch4 activity.

## Figures and Tables

**Figure 1 ijms-24-07502-f001:**
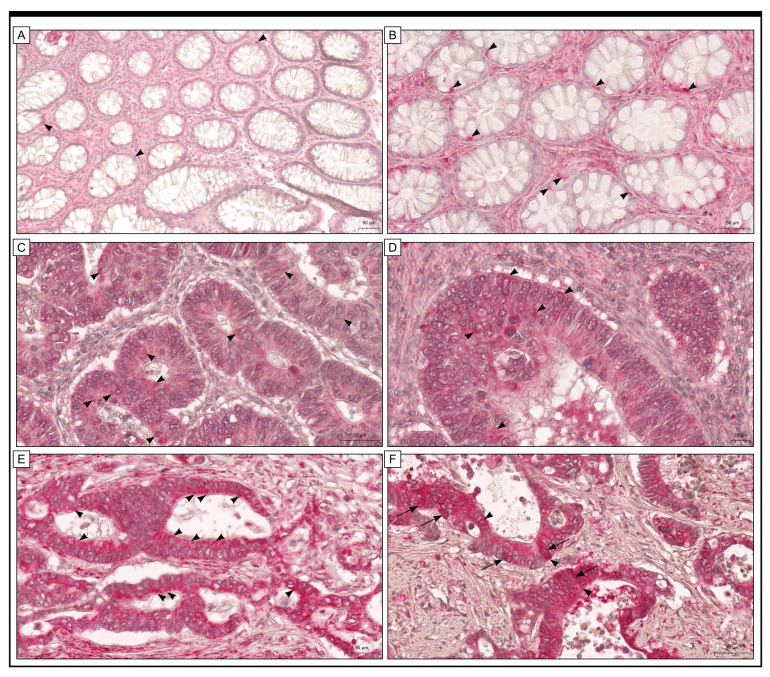
Representative microphotographs of immunohistochemical expression of Notch4 in colon adenocarcinoma tissue (**C**–**F**) and tissue margins with no cancerous lesions (adjacent non-tumour tissue) (**A**,**B**). (**A**,**B**) A low level of immunohistochemical reaction was detected in non-neoplastic cells. (**C**,**D**)—tumour presenting low Notch4 immunopositivity; (**E**,**F**)—tumour showing high Notch4 expression. The immunohistochemical expressions of Notch4 (red colour) are marked with arrowheads (expression in the cytoplasm) and arrows (expression in cell nuclei). Mayer’s haematoxylin was used to counterstain the nuclei (violet colour). The scale bar is 50 µm (**A**–**F**).

**Figure 2 ijms-24-07502-f002:**
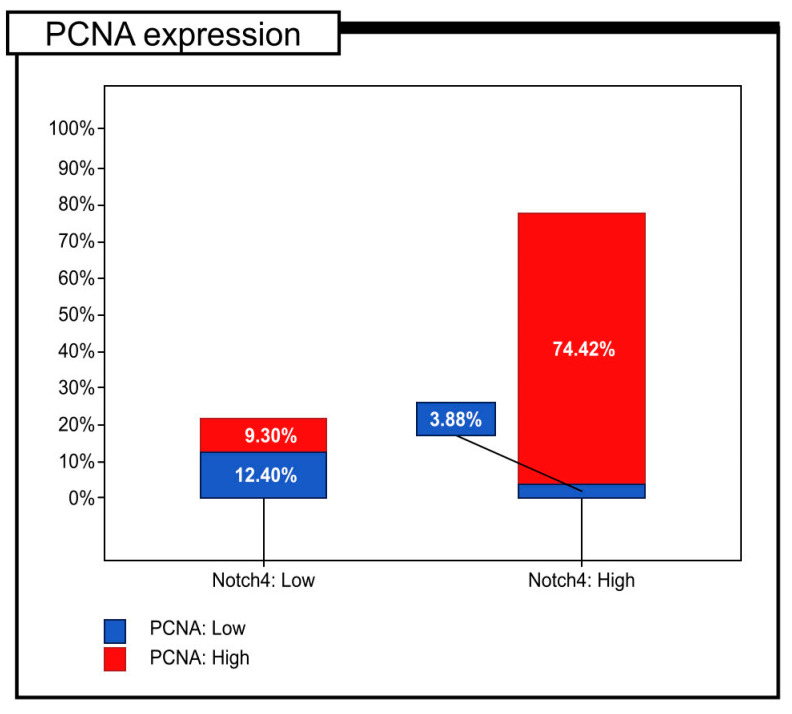
Percentage of immunohistochemical expression of PCNA defined as high and low expression in colon adenocarcinoma patients (*n* = 129).

**Figure 3 ijms-24-07502-f003:**
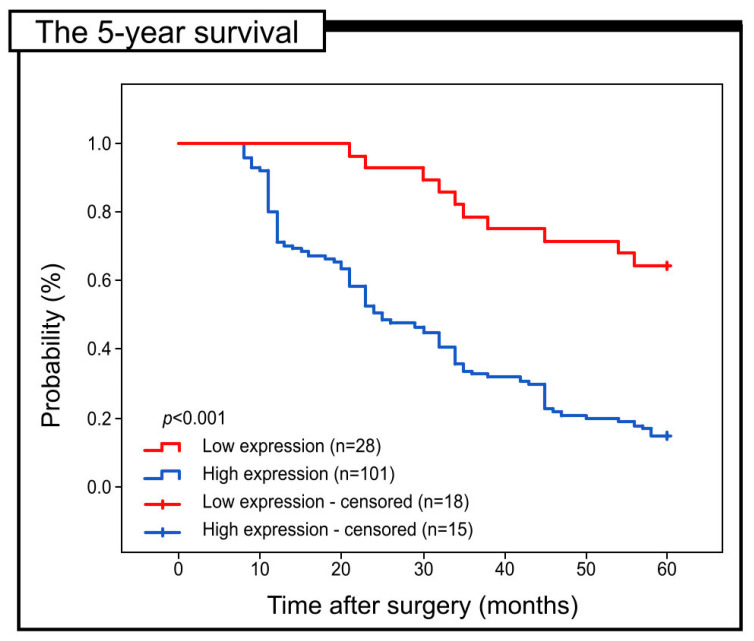
Kaplan–Meier curves of univariate analysis data (log-rank test) showing the 5-year survival rate for patients with high versus low Notch4 immunohistochemical expression.

**Figure 4 ijms-24-07502-f004:**
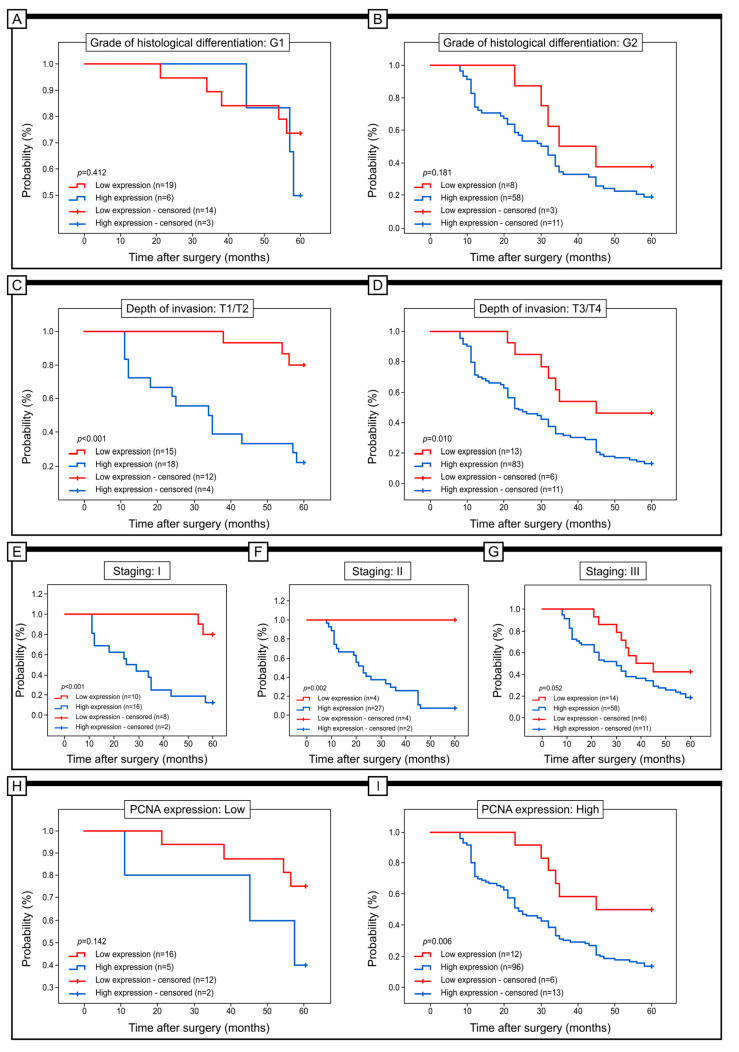
Kaplan- Meier curves of univariate analysis data (log-rank test) of patients with high versus low levels of Notch4 immunohistochemical expression. (**A**,**B**) The 5-year survival of patients with G1 (**A**) and G2 (**B**) grade of differentiation; with T1/T2 (**C**) and T3/T4 (**D**) depth of invasion; with staging I (**E**), staging II (**F**) and staging III (**G**); low level of immunohistochemical expression of PCNA (**H**) and high level of immunohistochemical expression of PCNA (**I**).

**Figure 5 ijms-24-07502-f005:**
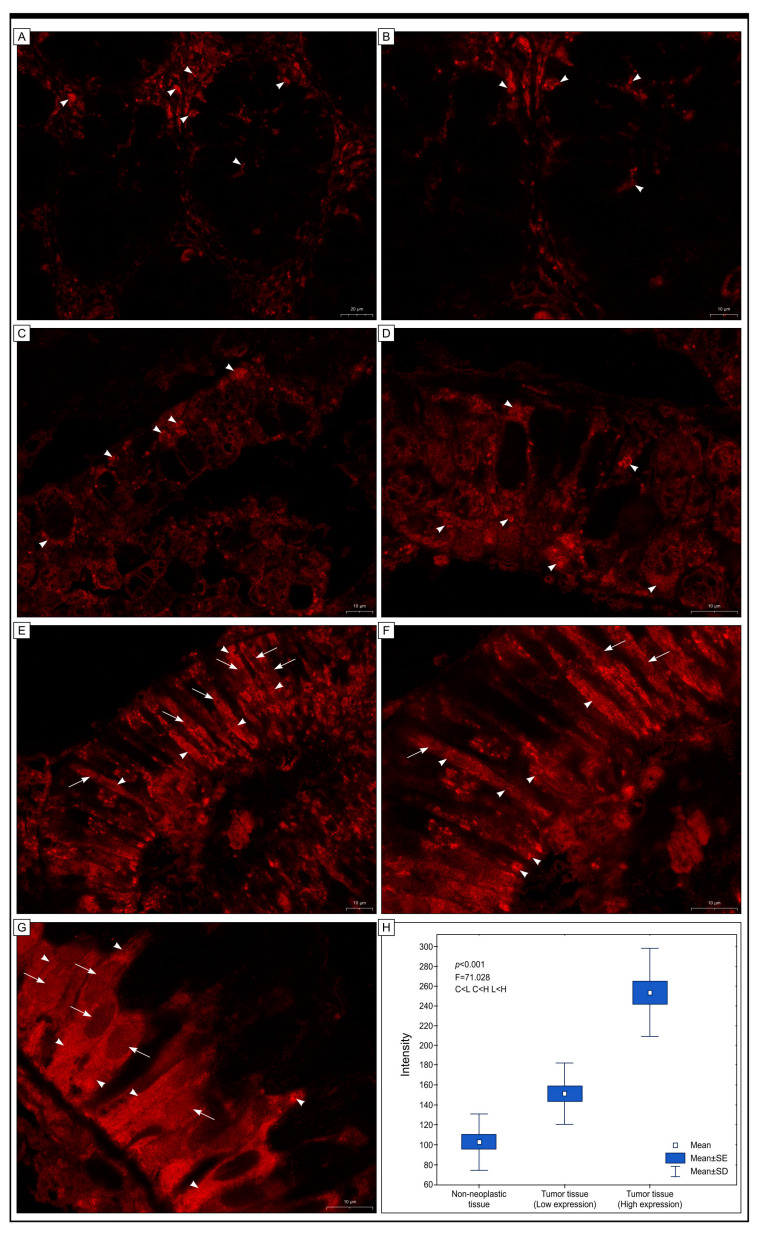
Representative microphotographs of Notch4 expression in colon adenocarcinoma tissue (**C**–**G**) and tissue margins with no cancerous lesions (adjacent non-tumour tissue) (**A**,**B**) by immunofluorescence. A fluorescent signal (red colour signal) of varying intensity was found in cells of non-neoplastic mucosa (**A**,**B**)—arrowheads indicate expression in the cytoplasm of non-pathological colonocytes) and cancer cells (**C**–**G**). In some cancer cells, the expression and fluorescence signal was found in the cytoplasm of the apical parts of the cells, while in others, intense fluorescence was found throughout all cytoplasm (arrowheads) of the cells or in the cell nuclei (arrows). (**H**)—results of ANOVA test showing differences between intensity of red signal indicating the presence of Notch among the tested groups; (**F**)—a result of ANOVA test; C < L, C < H, L < H—differences in intensity between the groups; C—non-neoplastic colon tissue, L—adenocarcinoma specimens with low expression of Notch4, H—adenocarcinoma specimens with high expression of Notch4.

**Figure 6 ijms-24-07502-f006:**
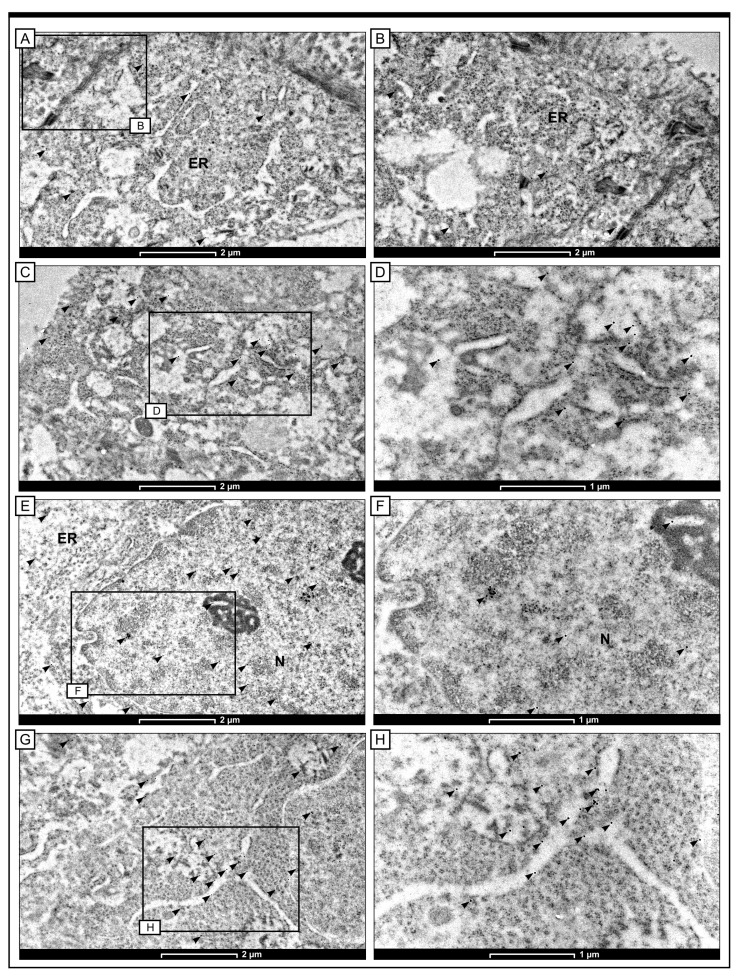
Immunogold labelling of Notch4 protein in cells within colon adenocarcinoma tissue. The small, black and electron-dense granules (arrowheads) were detected within the cytoplasm of colon epithelial cells in non-pathological colon mucosae (**A**,**B**). In cancer cells of adenocarcinoma specimens, the black granules were detected within the cytoplasm of the cells in mitochondria (**H**) and endoplasmic reticulum (**C**,**D**,**G**). Moreover, the electron-dense granules were also localised within the nuclei (**E**,**F**). N—cell nucleus; ER—endoplasmic reticulum.

**Table 1 ijms-24-07502-t001:** Characteristics of patients included in the study (*n* = 129).

		N (Number of Cases)	%
Gender	Females	62	48.06
	Males	67	51.94
Age [years]	≤60 years	44	34.11
	61–75 years	47	36.43
	>75 years	38	29.46
Grade of histological differentiation	G1	25	19.38
	G2	66	51.16
	G3	38	29.46
Depth of invasion	T1	15	11.63
	T2	18	13.95
	T3	73	56.59
	T4	23	17.83
Regional Lymph Node involvement	N0	57	44.18
	N1	38	29.46
	N2	34	26.36
Location of tumour	Right-sided tumours	66	51.16
	Left-sided tumours	63	48.84
Notch4 expression in healthy tissue	Low	118	91.47
	High	11	8.53
Angioinvasion	No	30	23.26
	Yes	99	76.74
PCNA expression	Low	21	16.28
	High	108	83.72
Staging	I	26	20.16
	II	31	24.03
	III	72	55.81

**Table 2 ijms-24-07502-t002:** Correlations between the expression of Notch4 protein and PCNA protein.

		The Immunoexpression Level of Notch4				*p*-Value
		Low		High		
The immunoexpression level of PCNA	Low	16	(12.40%)	5	(3.88%)	*p* < 0.001
	High	12	(9.30%)	96	(74.42%)	*p* = 0.146

**Table 3 ijms-24-07502-t003:** Correlations between the expression of Notch4 protein and clinicopathological characteristics in colon adenocarcinoma patients.

		The Immunoexpression Level of Notch4				*p*-Value
		Low		High		
Age [Years]	≤60 years	8	(18.18%)	36	(81.82%)	*p* = 0.670
	61–75 years	10	(21.28%)	37	(78.72%)	
	>75 years	10	(26.32%)	28	(73.68%)	
Gender	Females	15	(24.19%)	47	(75.81%)	*p* = 0.510
	Males	13	(19.40%)	54	(80.60%)	
Grade of histological differentiation	G1	19	(76.00%)	6	(24.00%)	*p* < 0.001
	G2	8	(12.12%)	58	(87.88%)	
	G3	1	(2.63%)	37	(97.37%)	
Depth of invasion	T1/T2	15	(45.45%)	18	(54.55%)	*p* < 0.001
	T3/T4	13	(13.54%)	83	(86.46%)	
Regional Lymph Node involvement	N0	14	(24.56%)	43	(75.44%)	*p* = 0.484
	N1/N2	14	(19.44%)	58	(80.56%)	
Localisation	Left-sided tumours	11	(16.67%)	55	(83.33%)	*p* = 0.155
	Right-sided tumours	17	(26.98%)	46	(73.02%)	
Angioinvasion	Yes	18	(60.00%)	12	(40.00%)	*p* < 0.001
	No	10	(10.10%)	89	(89.90%)	
PCNA expression	Low	16	(76.19%)	5	(23.81%)	*p* < 0.001
	High	12	(11.11%)	96	(88.89%)	
Staging	I	10	(38.46%)	16	(61.54%)	*p* = 0.052
	II	14	(19.44%)	58	(80.56%)	
	III	4	(12.90%)	27	(87.10%)	

**Table 4 ijms-24-07502-t004:** Univariate and multivariate analyses of various prognostic parameters in colon adenocarcinoma patients using Cox regression analyses.

Prognostic Parameter	Univariate Analysis			Multivariate Analysis		
	HR	95% CI	*p*-Value	HR	95% CI	*p*-Value
Gender	0.883	0.591–1.318	0.543	–	–	–
Age	0.999	0.985–1.014	0.929	–	–	–
Staging	1.152	0.896–1.481	0.269	–	–	–
Histological differentiation	2.415	1.785–3.267	<0.001	1.907	1.251–2.907	0.003
Depth of invasion	1.602	1.251–2.052	<0.001	0.985	0.715–1.358	0.928
Regional Lymph Node involvement	1.108	0.869–1.412	0.409	–	–	–
Localisation	1.091	0.731–1.629	0.668	–	–	–
Immunohistochemical expression of Notch4 in cancer tissue	4.389	2.269–8.487	<0.001	2.414	1.134–5.141	0.022
Angioinvasion	2.865	1.618–5.074	<0.001	0.896	0.427–1.880	0.772
PCNA expression	4.527	2.087–9.820	<0.001	1.454	0.486–4.355	0.503

## Data Availability

All data generated or analysed during this study are included in this article. Further, inquiries can be directed to the corresponding author.

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
