# Peer review of "The Clinical Application of Immunohistochemical Expression of Notch4 Protein in Patients with Colon Adenocarcinoma"

_ijms, 2023, doi:10.3390/ijms24087502_

Round 1
Reviewer 1 Report
The authors, based on a retrospective study, conclude that high Notch4 expression is correlated with a poor prognosis for patients with colon adenocarcinoma. The article is interesting and authors test 3 different techniques to analyze nocth4 expression.
There are, however, several concerns
Abstract:
- It is not mentioned that authors also performed immunofluorescence and TEM.
- The dilution used for notch4 antibody don’t need to be mentioned in the abstract
- It should be mentioned in the abstract and in the results that the study was performed on 129 colon adenocarcinomas.
- Line 25: maybe poor prognosis is more appropriate than worse prognosis
Introduction:
- Line 32: I’m not sure to understand what authors mean with “second most common malignancy globally” authors sould rephrase and add a reference (it is not the second for men and women based on American cancer society)
- Line 44: overall survival is cited for the first time, OS abbreviation has to be explained
Method:
4.1
- There is no explanation of grade 1, 2 and 3 that are used in the result section, it should be added. Same for T1/T2 invasion.
4.2
- Did authors confirm their method to stage notch4 with a 3rd pathologist? The quantification seems complicated? Did they check if the intensity alone or the % of positive cells alone was enough for the staging?
- Authors explain in introduction that notch4 is a receptor but IHC and immunofluorescence show cytoplasmic localization. Authors should explain which epitope is targeted by the antibody used. Did authors check with an antibody targeting another epitope?
- Line 327: remove the ) after nuclei
4.3 Statistical analysis:
- Authors should explain reasons why the use either chi2 or chi2 yatesa test.
4.4
- Is it the same reference of notch4 antibody used for IHC?
- I’m notch sure I understand well this paragraph: authors used Dako Liquid Permament Red, which according to agilent website, is a chromogen, to observe fluorescence? Therefore how could there observe fluorescence with a chromogenic technic? This technic must be performed again using real immunofluorescence antibody.
- If dako liquid permanent red is different from the one used for IHC the reference should be added.
- Which software was used for immunofluorescence imaging? This should be added.
4.5
- Line 358: authors should define LR in ‘LR white mixture (and add the reference).
- Is it the same notch4 antibody used as for IHC and IF? If not authors should add the reference
- authors should show images of controls in supplemental data.
Results:
2.1
- It should be mentioned that the study included 129 patients
- authors should explain G1/G2/G3 in the M&M section
- In pdf figures are in poor quality, therefore we can’t differentiate cytoplasmic and nuclear expression of notch4. Authors should add in figure 1 pictures with higher magnification that allow to visualize cytoplasmic and nuclear staining.
- It is not clear with the explanations given in introduction, where notch4 signal should be seen? At the plasma membrane? Golgi apparatus? Nucleus? This should be clarified.
2.2
- “Based on the results from IHC correlated the immunohistochemical status of Notch4 with the clinicopathological features of patients and the 5-year survival rate.” the sentence is not well written
- Line 110: Gpx-2 was never mentioned before. What is this protein? Why was it tested? How was it tested? It is not described in the M&M section…
- Authors should explain in the results why they decided to analyze PCNA
- Table2: PCNA expression/the immunoexpression level of Notch4: PCNA and notch4 are both performed by IHC with a qualitative result, why the titles are different?
- Line 144: G3 can’t be analysed for notch4 as the low expression contain only one patient
- Line 154: how can authors explain results on high level of PCNA expression as PCNA represent cells that are dividing, therefore more aggressive cancer?
2.4
- It is interesting to confirm notch4 expression with another technique than IHC. However this paragraph lack of interpretation. There is no result of the qualitative analysis, which software was used to nalyse the fluorescence intensity? on how many patients this technique has been used
I don’t see in figure 5 cells with nuclear fluorescence. Same remark as for figure 1, authors have to add some pictures with a higher magnification where we can see fluorescence in 1/ the cytoplasm with the apical part of the cell and , 2/all cytoplasm, 3/ the nucleus. Figures need more details.
- authors should correct the legend line 181: Notch4 expression of in colon
2.5
- It is interesting to show another technique and TEM has the advantage to identify the location within cells.
- How did authors differentiate fibroblasts from tumour cells? Did they selected areas before TEM? on how many patients this technique has been used?
- In figure 6 D, notch4 seems to be in the mitochondria? Any explanation why?
- Figure 6 legend: B is written twice and D is missing.
General remark:
Have authors checked on the expression of notch4 target genes (Hes and Hey family)? By RT-qPCR?
Discussion
- In the discussion, the localization of notch4 should be more discussed (no localization at the plasma membrane with immunogold? Is it a normal result in non-tumoral cells as fibroblasts?).
- Authors performed 3 different techniques to analyze Notch4 expression and localization but did not compare their results between techniques. It needs to be addressed.
- Line 272: abbreviations are used without definition
- Line 277: correct nfkb
- As notch4 seems to be implicated in EMT, did authors check on EMT markers as E-cadherin or twist for example?
Reviewer 2 Report
Comments on ijms-2283128-peer-review-v1 entitled “The clinical application of immunohistochemical expression of Notch4 protein in patients with colon adenocarcinoma” by Brzozowa-Zasada1et al.
The manuscript is an interesting study on the expression of Notch4 in colon adenocarcinoma and adjacent non-neoplastic tissue, and the association of this protein immunoexpression and classical clinicopathological features and overall survival. However, in my opinion, it is not necessary to perform immunohistochemistry and immunofluorescence to evaluate the expression Notch4, as the unique difference of these two techniques is the revelation system (a chromogen or a fluorochrome) and the microscope used (an optical or a fluorescence one). Even immunogold is not essential to the study, although it has the advantage of locate the protein in the cell compartments. Besides this criticism, I think this is a very complete study on the theme. However, there are some minor changes that should be amended to increase comprehensibility.
MINOR CHANGES
INTRODUCTION SECTION
Line 46: Please replace “novel biomarker” to “novel biomarkers”
Line 67: In the sentence “carcinoma (NSCLC)[26-28].” There´s a space missing. Please amend.
RESULTS SECTION
Please increase the space between table 1 and text (line 87).
Figure 1: Images should be replaced by new images with more nuclear contrast: the hematoxylin staining of nuclei is too faint, and nuclei are not clearly observed.
Lines 96-97-Figure 1 Legend- In the legend (and also in all manuscript) replace “healthy cells” to “non-neoplastic” cells. In fact, these cells on the periphery of tumor are influenced by the tumour microenviroment and may have some molecular/genetic changes…
Lines 97-98: Remove “patients” from the legend. Amend the sentence removing information about the patients with high or low expression (“…some patients…”;”…most patients…” ); this information is in the manuscript, and is not necessary to be insert in legends. An example of legends: “C, D- Tumor presenting low Notch4 immunopositivity; (E, F)- Tumor showing high Notch4 expression.”
Line 104- Amend the sentence …“Based on the results from IHC correlated the immunohistochemical status of Notch4 with…” to “The immunohistochemical status of Notch4 was correlated with…”
Line 109-110- Regarding the sentence …”The low level of immunohistochemical expression of Gpx-2 protein was found in 19 (76.00%)…” I´m not sure if authors present, previously in the manuscript, any information, about Gpx-2 protein. Please clarify this issue.
Table 2: Why there are some numbers in bold?? Please amend.
Line 120-122: Regarding the sentence “High Notch4 immunohistochemical expression was found in 89 (89.90%) of the patients with positive angioinvasion, while low immunoreactivity was 121 found in 10 (10.10%) patients”… I´m not sure if it is in accordance with data present in table 3: according to the table, 89,9% of patients with high Notch4 expression did not show lymphovascular invasion. Please confirm.
Line 150-152: I think authors should simplify the sentence; I suggest to substitute the phrase: “In patients with stage III, low expression of Notch4 demonstrated the association with a 5-year survival rate as well. However, in this case, the results were not statistically significant (log rank test, p= 0.052). “ to “In patients with stage III, low expression of Notch4 was also associated with a 5-year survival rate, although these results were not statistically significant (log rank test, p= 0.052).
Line 164: Replace “histological differentiation grade” to “histological grade”, as it was already mentioned that grading is performed according the degree of tumor/cell differentiation.
Lines 183-184: From the figure 5 legend remove …”by the use of confocal fluorescent microscope.” and replace it to “by immunofluorescence”. Remove “TexRed fluorescence (red signal) representing Notch4 protein was visualized with 592 nm excitation and 574- 735 nm emission filter sets.” as this information is already mentioned in Material and Methods section
Line 187-188: Please insert information regarding location of expression of Notch4 in non-neoplastic cells from surgical margins. In deed just in figure 6 legend, we verify that immunogold technique was performed in non-neoplastic tissue. Besides this, please also insert this information in material and methods section (lines 355-356), as in this section, authors just say “tissue samples” and do not mention if it is “neoplastic or non-neoplastic tissue”.
Line 190: Please amend: “stromal cells, especially fibroblasts. In Fibroblasts…“ to “stromal cells, especially fibroblasts. In fibroblasts…”
DISCUSSION SECTION
Line 246: I´m not sure that it is “100% correct” that PCNA is a “specific marker of cell division” as this marker is also associated to DNA repair mechanisms” (see Bologna-Molina et al 18(2):e174-e179 2013; or Lu et al 19, 233, 2019). Actually, Ki-67 is a better marker of proliferation than PCNA and probably due to an overestimation of the proliferating population, authors do not found association between survival and PCNA.
Line 264 and 266- There are some spaces missing.
Line 270 and 273- Please clarify the sentences: “In patients with NSCLC Notch4 expression was positively associated with tumour size, LNM, DM, and TNM stage. Patients with a high level of this protein had significantly lower OS than did Notch4-patients [26–28].” If I understand these sentences, “this protein” reports to “Notch4”, so the second phrase in confusing.
MATERIAL AND METHODS
Lines 316-340, 4.2. Immunohistochemical staining: Please insert the immunoreactivity method authors used to evaluate PCNA. In fact, authors just report the evaluation system used for Notch4.
Line 318: Please amend the sentence: “were cut into 4- m- thick sections”. There a “micro symbol” missing.
Line 327: In the sentence “…counterstain the nuclei).” There´s an extra parenthesis present.
Lines 330 and 332: Replace “frequency” to “extension”.
Line 334: Please Replace “…1, positive staining in 10 25% cells, …” to “…1, positive staining in 10-25% cells, …”
Line 341: 4.3. Statistical analysis: This item should be presented in the end of Material and methods section.
Line 349-353: 4.4. Confocal Fluorescent Microscopy- Please insert information regarding the immunofluorescence technique. As IHC and immunofluorescence techniques are similar, this 4.4 section could be merged with point 4.2 using the title “Immunohistochemical and immunofluorescence”. However, if authors intend to maintain this item, the title should be “immunofluorescence” instead of “Confocal Fluorescent Microscopy”
Lines 329-338: Please insert the methodology used to evaluate PCNA. In table 1 authors mention “high and low expression of PCNA but do not mention how they classified this marker, nor the cut-off value they used. If was a quantitative or a semiquantitative evaluation system. Please clarify.
Round 2
Reviewer 1 Report
lines 82-83 : do authors mean intracellular and intranuclear?
MET supplementary images : a legend must be added with comments of images.
Author Response
Once again, we would like to thank the reviewer for his valuable comments and insightful analysis.
lines 82-83 : do authors mean intracellular and intranuclear?
We apologise for this mistake. Of course, we had in mind intranuclear and intranucleolar localization according to the cited literature (Saini et al.).
MET supplementary images : a legend must be added with comments of images
Thank you for this comment. A figure with a description has been added to the supplementary material.